Expression profiles and gene set enrichment analysis of the transcriptomes from the cancer tissue, white adipose tissue and paracancer tissue with colorectal cancer

Zhang Xiufeng 1
Zhu Rui 2
Jiao Ye 3
Simayi Halizere 3
He Jialing 4
Shen Zhong 4
Wang Houdong 4
He Jun 4
Zhang Suzhan Zrsj@zju.edu.cn 1
Yang Fei yangfei919@zju.edu.cn 5
1 Department of Colorectal Surgery and Oncology (Key Laboratory of Cancer Prevention and Intervention, China National Ministry of Education), The Second Affiliated Hospital, Zhejiang University School of Medicine; Zhejiang Provincial Clinical Research Center, Cancer Center of Zhejiang University , Hangzhou , Zhejiang , China
2 Affiliated XiaoShan Hospital, Hangzhou Normal University , Hangzhou , Zhejiang , China
3 Chronic Disease Research Institute, The Children’s Hospital, and National Clinical Research Center for Child Health, School of Public Health, Zhejiang University , Hangzhou , Zhejiang , China
4 Department of Colorectal Surgery, Affiliated Hangzhou Dermatology Hospital, Zhejiang University School of Medicine , Hangzhou , Zhejiang , China
5 Department of Nutrition and Food Hygiene, School of Public Health, School of Medicine, Zhejiang University , Hangzhou , Zhejiang , China
Qin Jiangjiang
Electronic publication date: 2024 Mar 29
Publication date: 2024
Volume: 12
Electronic Location ID: e17105
Received 2023 Dec 19; Accepted 2024 Feb 22
Copyright: ©2024 Zhang et al.
Copyright year: 2024
Copyright holder: Zhang et al.
License: This is an open access article distributed under the terms of the Creative Commons Attribution License, which permits unrestricted use, distribution, reproduction and adaptation in any medium and for any purpose provided that it is properly attributed. For attribution, the original author(s), title, publication source (PeerJ) and either DOI or URL of the article must be cited.
License URL: https://creativecommons.org/licenses/by/4.0/

Keywords: CRC, WAT, PCT, Transcriptomes, lncRNA

Funding: National Natural Science Foundation 31770979 Pre-research Foundation of Zhejiang University High Sci-Tech Development Fund of Zhejiang University 2020QN026 The study was funded by the general program of National Natural Science Foundation (31770979 to Fei Yang); the Pre-research Foundation of Zhejiang University (2021) to Zhong Shen; the Pre-research Foundation of Zhejiang University (2022) to Xiufeng Zhang; High Sci-Tech Development Fund of Zhejiang University (2020QN026 to Fei Yang). The funders had no role in study design, data collection and analysis, decision to publish, or preparation of the manuscript.

==============================
Background

Colorectal cancer (CRC) is one of the most common cancers worldwide and is related to diet and obesity. Currently, crosstalk between lipid metabolism and CRC has been reported; however, the specific mechanism is not yet understood. In this study, we screened differentially expressed long non-coding RNAs (lncRNAs) and mRNAs from primary cancer, paracancer, and white adipose tissue of CRC patients. We screened and analyzed the genes differentially expressed between primary and paracancer tissue and between paracancer and white adipose tissue but not between primary and white adipose tissue. According to the results of the biological analysis, we speculated a lncRNA (MIR503HG) that may be involved in the crosstalk between CRC and lipid metabolism through exosome delivery.

Methods

We screened differentially expressed long non-coding RNAs (lncRNAs) and mRNAs from primary cancer, paracancer, and white adipose tissue of CRC patients. We screened and analyzed the genes differentially expressed between primary and paracancer tissue and between paracancer and white adipose tissue but not between primary and white adipose tissue.

Results

We speculated a lncRNA (MIR503HG) that may be involved in the crosstalk between CRC and lipid metabolism through exosome delivery.

Conclusions

In this study, the findings raise the possibility of crosstalk between lipid metabolism and CRC through the exosomal delivery of lncRNAs.

Introduction

Colorectal cancer (CRC) is one of the most common malignant tumors threatening human health worldwide, with over 1,900,000 cases and 935,000 deaths in 2020 (Sung et al., 2021). Currently, the primary CRC treatment for colorectal cancer includes surgery, chemoradiotherapy, and molecular targeted therapy. However, CRC is fundamentally difficult to cure, and despite advancements in screening and treatment, it remains a significant public health concern (Biller & Schrag, 2021).

Long non-coding RNAs (lncRNAs) are a class of RNA molecules over 200 nucleotides long, lacking protein-coding potential (Rinn & Chang, 2012). In recent years, the role of lncRNAs in CRC has garnered increasing attention, with studies suggesting they have a crucial role in the tumorigenesis and progression of CRC. By regulating gene expression and modification, lncRNAs are involved in various signaling pathways, such as Wnt/β-catenin, PI3K/Akt, and EMT, which affect the proliferation, metastasis, and prognosis of CRC (Duan et al., 2020; Zhang et al., 2021; Zhang et al., 2018).

The tumorigenesis of cancer is regulated by the tumor microenvironment (TME) (Bagaev et al., 2021). Cancer cells require a large amount of energy to grow and proliferate; their competition with neighboring tissue for substances can cause metabolic abnormalities in those tissue. Furthermore, secretions produced by the metabolic activities of the TME lead to the metabolic activity of the surrounding cells (O’Sullivan et al., 2019a). White adipose tissue (WAT) is a vital component of TME in colorectal cancer; WAT is also important in energy storage and the regulation of systemic metabolic homeostasis, and WAT browning is involved in cachexia in CRC (Hildreth et al., 2021). The lipid microenvironment around cancer cells is associated with CRC tumorigenesis, progression, and metastasis (Castellano-Castillo et al., 2019). Peritumoral adipose tissue affects adipocytes, as well as immune and endothelial cells (Tabuso et al., 2021). Therefore, among adipose metabolism disorders, a direct and independent relationship has been ascertained for some cancers, including CRC. However, the specific regulatory mechanisms underlying lipid metabolism and CRC progression remain poorly understood. In this study, we analyzed the lncRNA and mRNA profiles in the CRC tissue, WAT, and paracancer tissue (PCT) of CRC patients using high-throughput RNA sequencing and functional prediction analysis to explore the crosstalk mechanism between CRC tissue and WAT and confirm the importance of the CRC microenvironment and propose a novel therapeutic treatment for CRC.

Materials & Methods

Tissue specimen collection

A total of 10 CRC patients were involved in this study. An experienced clinician removed WAT specimens from the mesentery 5 cm adjacent to the cancer or PCT of each patient. The collected tissue was rinsed with saline at 4 °C to remove bacteria and blood attached to the surface and immediately stored at −80 °C before sequencing and analysis. This study was approved by the Ethics Committee of the Affiliated Hangzhou Dermatology Hospital, Zhejiang University School of Medicine, and informed consent was obtained from all participants (No. 2021KA034). All patients participating in the study have signed informed consent.

High-throughput sequencing

We used TRIzol reagent (Invitrogen, Carlsbad, CA, USA) to extract RNA from primary cancer tissue, PCT, and WAT. The RNA of all samples with the Ribo-ZeroTM Magnetic Kit (Epicentre, San Diego, CA, USA) was fragmented using First Strand Master Mix (Invitrogen, Carlsbad, CA, USA). The first strand cDNA was generated by reverse transcription using random primers, after which the second strand cDNA was synthesized, the RNA template was elimated, and dTTP was replaced by dUTP. The linked products were amplified and purified. We completed RNA sequencing library construction by using BGI (Wuhan, Hubei, China).

Prediction of subcellular lncRNA Localization

We obtained lncRNA sequences from Dr. Tom Data Visualization (BGI).The lncRNA subcellular localization was predicted using input transcripts in lncLocator (Shanghai Jiao Tong University, http://www.csbio.sjtu.edu.cn/bioinf/lncLocator/) (Cao et al., 2018). GraphPad Prism 9 software (version 8.0; GraphPad Software, San Diego, CA, USA) was used to show and analyze the result.

Gene set enrichment analysis

Gene set enrichment analysis (GSEA) was performed using the Dr. Tom system with the Broad Institute software package (http://biosys.bgi.com, BGI), based on the Kyoto Encyclopedia of Gene and Genome (KEGG) pathway database.

Protein–protein interaction network analysis

To illustrate the functional interactions of the screened mRNAs, the Dr. Tom system was used to construct and visualize a protein–protein interaction (PPI) network. The scores ranged from 0 to 1,000. Significant interactions were considered with a required combined score of >950.

Construction of the mRNA–lncRNA co-expression network

The Pearson coefficient (r) >0.9 and p < 0.05 mRNA-LncRNA pairs were used to construct the co-expression network, which was visualized using the Cytoscape software (version 3.8.0, San Francisco, California, USA).

Gene ontology and KEGG pathway analysis

We further analyzed the potential biological processes, cellular components, molecular functions, and pathways of the target genes. The Dr. Tom system included Gene Ontology (GO) and KEGG pathway enrichment analyses. Predominately enriched GO terms were defined as those with a p-value < 0.05. A heatmap was created and enriched using R studio.

Results

Expression profiles and GSEA of CRC and PCT transcriptomes

We obtained tissue samples from CRC tissue, PCT, and WAT and performed high-throughput sequencing. In order to avoid error caused by taking samples, we took CRC from the primary cancer, PCLT and WAT were taken from the normal tissue and mesangial fat, which was 5 cm from the edge of primary cancer (Fig. 1A). We identified 19,514 and 107,009 genes as mRNAs and lncRNAs, respectively. We identified 4,959 DEGs between WAT and CRC tissue (WAT/CRC), 2,848 between PCT and WAT, and 1,509 between CRC tissue and PCT (CRC/PCT).

Figure 1 Expression profiles and GSEA of CRC and PCT transcriptomes.

(A) Colorectal cancer, (B) paracancer tissue, (C) white adipose tissue.

We then used GSEA to observe the function of genes contained in CRC tissue and PCT. We used strict inclusion criteria (i.e., normalized enrichment score (NES) >1.0, nominal p-value <5%, FDR <25%). We found that 25 of the 162 detected gene sets were enriched in CRC tissue, compared to PCT. These included the ubiquitin-mediated proteolysis, Fanconi anemia pathway, steroid biosynthesis, p53 signaling pathway, and basal transcription factors. Table S1 shows the specific information of 26 significant enrichment pathways. These pathways are related to metabolism, genetic information processing, and environmental information processing. The alteration of steroid biosynthesis regulates lipid metabolism, and the change in local steroid concentration can cause abnormal adipose tissue function, such as dyslipogenesis or abnormal production of adipokines, eventually leading to lipid metabolism disorder (Ostinelli et al., 2022; Wawrzkiewicz-Jałowiecka, Lalik & Soveral, 2021). Twenty candidate genes of interest were enriched in steroid biosynthesis, and the leading edge included 14 genes (Fig. 1B). The p53 signaling pathway influences the tumorigenesis of CRC; TP53 mutations in CRC often affect the biological function of wild-type p53, such as promoting cancer cell stemness, proliferation, invasion, and metastasis, all of which contribute to cancer progression (Giacomelli et al., 2018). There were 73 candidate genes of interest enriched in the p53 signaling pathway, and the leading-edge subsets included 29 genes (Fig. 1C). In order to avoid error casued by taking samples, we took CRC from the primary cancer, PCLT and WAT were taken from the normal tissue and mesangial fat, which was 5 cm from the edge of primary cancer.

Protein–protein interaction network construction and analysis

To identify the key genes enriched in these two pathways and study the mechanism of crosstalk between lipid metabolism and CRC tumorigenesis, a network analysis of protein–protein interaction (PPI) was conducted. Genes with an overall interaction score of 950 (0–1,000) were considered important and likely core genes. Possible core genes in steroid biosynthesis and the p53 signaling pathway are shown in Figs. 2A and 2B (Table S2). Two functional clusters were constructed for the steroid biosynthesis and p53 signaling pathway. The KEGG pathway enrichment analysis of functional clusters in Fig. 2A show that genes in this functional cluster were involved in steroid synthesis, cholesterol metabolism, and parathyroid hormone synthesis (Fig. 2C). Figure 2B shows that the clusters were involved in the p53 signaling pathway and other cancer tumorigenesis-related pathways, including apoptosis (Fig. 2D).

Figure 2 Protein–protein interaction network construction and analysis.

Analysis of differential genes between CRC tissue and PCT

Using DESeq2 R package and setting strict inclusion criteria (log2 fold change ≥ 1, FDR ≤ 0.001), we obtained DEGs between CRC tissue and PCT from the RNA-seq database. We identified 1,228 genes with altered expression levels between CRC tissue and PCT (Table S3). Figure 3A depicts the genes differentially expressed between the CRC tissue and PCT. We found that, compared with PCT, 190 differentially expressed lncRNAs (DElncRNAs) and 435 differentially expressed mRNAs (DEmRNAs) were upregulated in CRC tissue, while 54 DElncRNAs and 191 DEmRNAs were downregulated. We visualized the top 80 DElncRNAs as expressive heatmaps using normalized raw z-score (Fig. 3B). Similarly, cluster analysis was performed on the top 100 DEmRNAs (Fig. 3C).

Figure 3 Analysis of differential genes between CRC tissue and PCT.

Numerous studies have highlighted the ability of lncRNAs to bind to microRNA (miRNA) sites as competing endogenous RNAs (ceRNAs), thereby affecting and regulating the expression of mRNAs and target genes (Xu et al., 2022). Based on the ceRNA network constructed by the Dr. Tom system, we identified 476 target mRNAs between CRC tissue and PCT and performed KEGG pathway enrichment analysis. The results indicated that the PPAR, TNF, NF-kappa B, MAPK, and NOD-like receptor signaling pathways were associated with the tumorigenesis of CRC and lipid metabolism (Fig. 3D).

To identify the potential functions of the DEGs between the CRC tissue and PCT, we analyzed 626 DEmRNAs using the KEGG pathway list. The results showed that the upregulated expression of 435 in CRC tissue resulted in significant enrichment of DEmRNA in 13 of the 255 pathways.The 13 KEGG pathways belong to the following level 1 categories: organic systems, cellular processes and environmental information processing, genetic information processing, metabolism, and human diseases. Table S4 shows the detailed information of the level 2 KEGG terms, including eukaryotic ribosome biogenesis, steroid biosynthesis, the IL-17 signaling pathway, cytokine-cytokine receptor interaction, homologous recombination, and nucleocytoplasmic transport. Among these 13 pathways, steroid biosynthesis was associated with lipid metabolism (Fig. 3E and Table S3). We found that 46 pathways were downregulated in CRC tissue (p < 0.05) (Table S4). According to the level 1 KEGG categories, we identified 20 pathways in organismal systems, cellular processes and environmental information processing, genetic information processing, metabolism, and human diseases. The level 2 KEGG terms include the regulation of lipolysis in adipocytes, which is associated with lipid metabolism. We also observed that the B cell receptor signaling pathway (Fig. 3F), which is related to immunity, critically regulates the tumorigenesis of cancer (Tanaka & Baba, 2020).

Analysis of DEGs between CRC tissue and PCT and between PCT and WAT, but not between WAT and CRC tissue

To further analyze the correlation between CRC progression and lipid metabolism, we selected 974 genes (log2 fold change >1, FDR <0.05) that revealed differences between CRC tissue and PCT and between PCT and WAT but not between WAT and CRC tissue. The 974 genes, 671 mRNAs, 214 lncRNAs, and other types of RNAs are presented in Table S7. We performed an expression clustering analysis (Fig. 4A). The KEGG pathway analysis revealed that the 671 mRNAs were associated with human diseases, organismal systems, environmental information processing, metabolism, cellular processes, and genetic formation processing (Fig. 4B). The GO analysis of the 671 mRNAs indicated that the genes were enriched in cytosolic ribosomes, large ribosomal subunits, large ribosomal subunits, and extracellular exosomes (Fig. 4C). When normal tissue become cancerous, they often produce more exosomes, and as exosomes are excreted, cancer cells change their immediately surrounding and distant microenvironment (Zhang & Yu, 2019). Therefore, we analyzed the genes enriched in the extracellular exosomes by using the KEGG pathway. The result suggested that the ECM-receptor interaction is related to the regulatory network and its prognostic role in CRC (Fig. 4E).

Figure 4 Analysis of DEGs between CRC tissue and PCT and between PCT and WAT, but not between WAT and CRC tissue.

Based on the Dr.Tom ceRNA network database, we identified 2,676 mRNAs regulated by 214 lncRNAs (Fig. 4D and Table S5). A KEGG pathway enrichment analysis was performed on the genes in the identified networks, and several pathways were predominately enriched. The results revealed that the HIF-1α signaling, arginine, and proline pathways enriched in the target genes were related to the tumorigenesis of several cancers, including CRC (Fig. 4F). The absolute values of log2 (CRC/PCT) and (WAT/PCT) were ≥ 2, and 75 lncRNAs were screened. To further clarify the differences in the expression levels of these lncRNAs among the groups, we performed a cluster analysis (Fig. 4G). According to these criteria, the expression levels of both WAT and CRC genes were higher than those of PCT. The following lncRNAs were screened: MIR222HG, DUXAP8, DUXAP9, LINC00920, MIR503HG, URS0001BF3125, DNAH17-AS1, lncRNA-ATB, HSD52, FAM222A-AS1, URS00008B328B, URS0000D584C2, URS0000D5BAAB, and URS0001BF711B. Based on several studies that found that the subcellular localization of lncRNA directly affects its function, we used lncLocator to analyze the locations of these 14 lncRNAs. Database prediction results indicated that lncRNA-ATB localized to the nucleus, MIR503HG localized to the exosomes, and the others localized to the cytoplasm or cytosol (Fig. 4H). We predicted the mRNA targeting MIR503HG based on the ceRNA network database and performed a GO analysis (Fig. 4I and Table S6).

Co-expression analysis and functional prediction

To further explore the interaction of MIR503HG with CRC progression and tumorigenesis, we determined the correlation between MIR503HG and 3,988 mRNA expression levels in the DEGs between the CRC tissue and PCT by calculating the Pearson correlation coefficient (r) (Fig. 5A). We also found that MIR503HG exhibited a strong correlation with 20 mRNAs (r > 0.9, Fig. 5B). We then performed KEGG pathway enrichment analysis to identify the genes with strong correlations for functional analysis. Genes co-expressed with MIR503HG were enriched in the PI3K-Akt, Rap1, and calcium signaling pathways (Fig. 5C).

Figure 5 The correlation between MIR503HG and 3988 mRNA expression levels in the DEGs between the CRC tissue and PCT.

Discussion

We aimed to study the crosstalk mechanism between lipid metabolism and cancer progression. Our RNA-seq data on CRC tissue, PCT, and WAT showed that lncRNA may facilitate substance exchange between CRC tissue and WAT through exosomes, thus affecting the TME.

Our findings revealed that steroid biosynthesis and p53 signaling were activated in CRC tissue. Enrichment analysis indicated that the genes enriched in the steroid pathway are also involved in cholesterol metabolism. White adipose tissue is an important site of cholesterol synthesis and metabolism, and multivesicular bodies (MVBs) formation is central to exosome biogenesis (Grodin, Siiteri & MacDonald, 1973). The early endosomes absorb plentiful goods into intracavitary vesicles (ILVs) to generate MVBs. Upon maturity, MVBs dynamically communicate with other organelles by various means (Rabas et al., 2021; Zhao, Codogno & Zhang, 2021). All these communications regulate MVB formation and ILV molecular composition. Eventually, the mature MVBs either fuse with lysosomes and are degraded or fuse with the plasma membrane to release ILVs, which are called exosomes. Cholesterol is enriched in the exosomes and involved in the localization and trafficking of late exosomes (Ciardiello et al., 2020; Ikonen & Zhou, 2021; Wilhelm et al., 2017). This suggests that lipid metabolism is altered during the tumorigenesis of CRC, thereby regulating the secretion of steroids and cholesterol.

The p53 signaling pathway is involved in the tumorigenesis of CRC. Loss of wild-type p53 function owing to gene mutations and other mechanisms, such as overexpression of negative p53 regulators, has been recognized as a prerequisite for the progression of many human cancers (Donehower et al., 2019). Simultaneously, p53 affects the secretion of exosomes, thus affecting the tumorigenesis of cancer (Azulay, Cooks & Elkabets, 2020; Domenis et al., 2021). Wild-type p53 increases exosome secretion by upregulating the transcription of its target genes, such as caveolin (Giacomelli et al., 2018). Lipid metabolism in cancer tissue may differ from that in normal tissue, which may regulate the tumorigenesis of CRC by affecting exosome synthesis.

We screened differentially expressed mRNAs and lncRNAs in both CRC tissue and PCT, and WAT and PCT, but not in CRC tissue and WAT. We performed KEGG pathway analysis on some DEmRNAs, focusing on the PI3K-Akt signaling pathway and ECM-receptor interaction immunodeficiency. The PI3K-Akt signaling pathway is related to the proliferation and metastasis of CRC via regulation of autophagy (Jiang et al., 2021; Ma, Lou & Jiang, 2020), and ECM-receptor interactions regulate liver metastasis in CRC (Machackova et al., 2020). Our GO analysis of the DEmRNAs revealed that 122 mRNAs were enriched in the exosomes. The KEGG pathway enrichment analysis of the predominantly enriched genes in the exosomes indicated that most pathways were related to energy metabolism.

Exosome-enriched genes affect ECM-receptor interaction. To determine the functional relevance of these 214 lncRNAs, we used the ceRNA database provided by Dr.Tom (https://biosys.bgi.com/) and predicted lncRNA as the target mRNA via ceRNA. We performed KEGG pathway enrichment analysis on these mRNAs and observed the HIF-1-α pathway. The HIF-1-α pathway mediates local hypoxia in tumor tissue; HIF1-α promotes the secretion of exosomes, but the specific mechanism is unclear (Muñiz García et al., 2022). Exosomes specifically target lncRNAs. By setting strict inclusion criteria, we selected a lncRNA (MIR503HG) that may be targeted by the exosomes between the CRC tissue and WAT. Based on the ceRNA network, we predicted the target mRNAs and performed a GO analysis, which revealed that they were enriched in the Smad pathway. The TGF-Smad signaling pathway has a role in promoting the proliferation, invasion, and migration of rectal cancer cells (Wang et al., 2019a; Wang et al., 2019b), and the expression of Smad family members is altered in CRC (Fleming et al., 2013; Jiang et al., 2019; Wu et al., 2021). We successfully screened exosomes encapsulating MIR503HG, and they had the specificity to target CRC and WAT.

The function of MIR503HG in cancer regulation was also investigated. We performed a correlation analysis between them and the differentially expressed genes between the CRC tissue and PCT and identified 20 strongly correlated genes (r > 0.9). The KEGG enrichment analysis of these strongly correlated genes showed that the pathways enriched by miR503HG co-expressed genes were related to multiple CRC progression and energy metabolism pathways. Among them, the disorder of the calcium signaling pathway is involved in the tumorigenesis of many cancers. The release or influx of calcium ions from inside and outside cells promotes exosome biogenesis under pathological and physiological conditions (Savina et al., 2005).

Currently, there is interest in the molecular crosstalk between cancer cells and adipocytes. Exosomes, cystic carriers with a diameter of 40–150 nm, have emerged as novel mediators of various diseases (Kalluri, 2016; Yu et al., 2022). Exosomes participate in intercellular communication between different cells by transferring molecules including miRNAs, lncRNAs, and proteins (Kalluri, 2016). Reducing the generation of exosomes can inhibit the tumorigenesis of CRC (Wang et al., 2019a; Wang et al., 2019b). We observed that CRC progression may be regulated by lipid metabolism through the tissue-to-tissue delivery of exosome-encapsulated MIR503HG; lipid metabolism may affect the progression of CRC. Our results suggest that crosstalk between CRC and WAT occurs via lncRNA-coated exosomes. We also speculated 20 mRNAs that might interact with lncRNAs to affect CRC progression.

Our study has some limitations. First, owing to the small number of samples, which limits the generalizability of the findings. Second, the impact of cancer tissue metastasis on data analysis was not considered during data collection.

Conclusions

We comprehensively analyzed the differentially expressed lncRNAs associated with lipid metabolism in CRC patients. The results indicated that exosomes are a possible pathway for material exchange in experimental CRC and adipose tissue. In addition, our data suggested that lncRNA MIR503HG, which mediates exosomal exchange between cancer and adipose tissues, may affect CRC progression by influencing the secretion of Smad family factors. In addition, basic experiments were conducted to further explore the specific mechanism of MIR503HG to achieve CRC tissue and WAT through exosome delivery. We will further increase the reliability of the data by expanding the sample data size in future studies.

Supplemental Information

Supplemental Information 1 Raw data

Table S1 The specific information of 26 significant enrichment pathways by GSEA to observe the function of genes contained in CRC tissue and PCT

Table S2 The possible core genes in steroid biosynthesis and the p53 signaling pathway between lipid metabolism and CRC tumorigenesis

Table S3 The DEGs between CRC tissue and PCT from the RNA-seq database

We identified 1228 genes with altered expression levels between CRC tissue and PCT.

Table S4 46 pathways were downregulated in CRC tissue (p ¡ 0.05) compared to PCT

Table S5 2676 mRNAs regulated by 214 lncRNAs from DEGs between CRC and PCT

Table S6 MIR503HG based on the ceRNA network database

Table S7 974 genes, 671 mRNAs, 214 lncRNAs, and other types of RNAs are presented differences between CRC tissue and PCT and between PCT and WAT but not between WAT and CRC tissue

Additional Information and Declarations

Competing Interests

Author Contributions

Human Ethics

Clinical Trial Ethics

DNA Deposition

Data Availability

Clinical Trial Registration

The authors declare there are no competing interests.

Xiufeng Zhang conceived and designed the experiments, performed the experiments, analyzed the data, prepared figures and/or tables, authored or reviewed drafts of the article, and approved the final draft.

Rui Zhu conceived and designed the experiments, performed the experiments, analyzed the data, prepared figures and/or tables, authored or reviewed drafts of the article, and approved the final draft.

Ye Jiao conceived and designed the experiments, performed the experiments, analyzed the data, prepared figures and/or tables, authored or reviewed drafts of the article, and approved the final draft.

Halizere Simayi conceived and designed the experiments, performed the experiments, analyzed the data, prepared figures and/or tables, authored or reviewed drafts of the article, and approved the final draft.

Jialing He conceived and designed the experiments, performed the experiments, analyzed the data, prepared figures and/or tables, authored or reviewed drafts of the article, and approved the final draft.

Zhong Shen conceived and designed the experiments, performed the experiments, analyzed the data, prepared figures and/or tables, authored or reviewed drafts of the article, and approved the final draft.

Houdong Wang conceived and designed the experiments, performed the experiments, analyzed the data, prepared figures and/or tables, authored or reviewed drafts of the article, and approved the final draft.

Jun He conceived and designed the experiments, performed the experiments, analyzed the data, prepared figures and/or tables, authored or reviewed drafts of the article, and approved the final draft.

Suzhan Zhang conceived and designed the experiments, performed the experiments, analyzed the data, prepared figures and/or tables, authored or reviewed drafts of the article, and approved the final draft.

Fei Yang conceived and designed the experiments, performed the experiments, analyzed the data, prepared figures and/or tables, authored or reviewed drafts of the article, and approved the final draft.

The following information was supplied relating to ethical approvals (i.e., approving body and any reference numbers):

This study was approved by the Ethics Committee of the Affiliated Hangzhou Dermatology Hospital, Zhejiang University School of Medicine(Ethical Application Ref: 2021KA034), and informed consent was obtained from all participants.

The following information was supplied relating to ethical approvals (i.e., approving body and any reference numbers):

This study was approved by the Ethics Committee of the Affiliated Hangzhou Dermatology Hospital, Zhejiang University School of Medicine, and informed consent was obtained from all participants.

The following information was supplied regarding the deposition of DNA sequences:

The group II intron/IEP sequences are available at GenBank: GSE249054.

The following information was supplied regarding data availability:

The data is available at NCBI GEO: GSE249054.

The following information was supplied regarding Clinical Trial registration:

No. 2021KA034

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
