# Peer review of "Expression profiles and gene set enrichment analysis of the transcriptomes from the cancer tissue, white adipose tissue and paracancer tissue with colorectal cancer"

_PeerJ, doi:10.7717/peerj.17105_

## Round 0.1 · original submission · Minor Revisions

Please carefully read the comments and suggestions from the reviewers. A point-by-point response and a revised manuscript should be provided.

Reviewer 1 ·

Basic reporting

No comment

Experimental design

No comment

Validity of the findings

No comment

Additional comments

1.Please provide basic information of the enrolled patients with colorectal cancers in your manuscript and could show it on a table (about age, gender,tumor location,pathological type,TNM staging,vascular cancer embolus, lymphatic metastasis and tumor budding) so as to make your research better understood.
2.The tumorigenesis of cancer is regulated by the tumor microenvironment.Can you elaborate further on the significance of the crosstalk analysis between the mesenteric fat and the tumor tissue?
3.The conclusion of the paper should be careful to avoid overstating the points that have not been fully verified or the future application prospects.

Reviewer 2 ·

Basic reporting

No comment

Experimental design

No comment

Validity of the findings

No comment

Additional comments

Colorectal cancer is closely related to diet and obesity. It’s very meaningful to compare the lncRNA level between colorectal cancer (CRC), white adipose tissue (WAT) and paracancer tissue (PCT), as they are from similar lineage.I suggest accepting this paper after the revision.Some comments are as follows:
1.please provide the baseline characteristics information of the patient including sample size for lncRNA expression analysis, BMI of the patient.
2.The authors need to add more discussion of LncRNA miR503HG.
3.Does the mesenteric fat involve in the tumor microenvironment?

·

Basic reporting

Comments:
At present, most studies have confirmed the correlation between colon cancer and lipid metabolism caused by obesity.In this paper, the researchers attempted to analyze the relationship between mesangial fat and the development of colon cancer. Overall, these are the novel observations, and the data support the main conclusion. However, there are minor concerns that need to be addressed, which are listed as follows:
1.Why was mesangial fat chosen as the study object instead of the fat around the tumor?
2. What is the relationship between mesangial fat and systemic lipid metabolism? Does mesangial fat reflect systemic lipid metabolism?
3. Line 60“from the mesentery 5 cm adjacent to the cancer or PCT of each patient”,Why choose 5cm?
4. Line 48-50,Please clarify the relationship between the mesentery and white adipose tissue.
5.Line 198-199,"Based on several studies ...". Please add relevant references.
6. Please provide the basic information of the histological sample patient, such as age, gender, tumor location, local invasion depth, lymph node metastasis, distant metastasis, pTNM stage, etc.
7.Do you think comparing the RNA level of WAT near the CRC tissue with WAT near the PCT makes more sense?
8.Please provide relevant ethical review material.
9. Please provide high-resolution figures. The quality of some figures are not good such as Figure 5g.
10. Some grammar errors have been identified. The authors should seek help from certified English Editing Service to polish its English.

Experimental design

The experimental design is ok.

Validity of the findings

Overall, the findings are convincing.

---

## Round 0.2 · accepted · Accept

The authors have addressed the questions from all reviewers.

Reviewer 1 ·

Basic reporting

No comment.

Experimental design

No comment.

Validity of the findings

No comment.

Additional comments

No comment.

Reviewer 2 ·

Basic reporting

No comment.

Experimental design

No comment.

Validity of the findings

No comment.

Additional comments

No comment.

·

Basic reporting

The authors have well addressed my concerns.

Experimental design

The authors have well addressed my concerns.

Validity of the findings

The authors have well addressed my concerns.